# A corset function of exoskeletal ECM promotes body elongation in *Drosophila*

Reiko Tajiri [1✉], Haruhiko Fujiwara[1] & Tetsuya Kojima [1✉]

Body elongation is a general feature of development. Postembryonically, the body needs to be framed and protected by extracellular materials, such as the skeleton, the skin and the shell, which have greater strength than cells. Thus, body elongation after embryogenesis must be reconciled with those rigid extracellular materials. Here we show that the exoskeleton (cuticle) coating the *Drosophila* larval body has a mechanical property to expand less efficiently along the body circumference than along the anteroposterior axis. This "corset" property of the cuticle directs a change in body shape during body growth from a relatively round shape to an elongated one. Furthermore, the corset property depends on the functions of Cuticular protein 11 A and Tubby, protein components of a sub-surface layer of the larval cuticle. Thus, constructing a stretchable cuticle and supplying it with components that confer circumferential stiffness is the fly's strategy for executing postembryonic body elongation.

[1] Department of Integrated Biosciences, Graduate School of Frontier Sciences, The University of Tokyo, Biosciences Building 501, 5-1-5 Kashiwa-no-ha, Kashiwa-shi, Chiba 277-8562, Japan. ✉email: rtajiri@edu.k.u-tokyo.ac.jp; tkojima@k.u-tokyo.ac.jp

Transformation of a relatively round-shaped body into an elongated one is a general feature of development. Embryonic axis elongation in a wide range of animals has been commonly explained by active and coordinated cell behaviors[1], such as oriented cell division[2,3], cell intercalation[4–8] and cell migration[9,10]. Postembryonic body elongation requires a different consideration: because materials that frame the body during postembryonic life, such as the skeleton, the skin and the shell, are mainly shaped by extracellular matrix (ECM), body shape change must involve shape change of those materials outside of cells.

The insect body is coated by cuticle, a seamless sheet of ECM composed of proteins and chitin (a linear polymer of N-acetyl-beta-D-glucosamine (GlcNAc)) that are deposited by the epidermis[11]. The cuticle is continuously required throughout postembryonic life as an external skeleton that gives the insect its shape, supports movement, and protects against injury and desiccation[11,12]. In the developmental context, the textbook view of the cuticle is that it restricts changes in body shape and size, and that it needs to be replaced through molting to allow for those changes[11,13,14]. Nonetheless, stretching of cuticle between molts has been shown to accompany body size increase in some insects[15–18]. Furthermore, ECM dynamics has emerged as a general mechanism of insect morphogenesis[19]. For example, deformation of the cuticle covering the larval body of *Drosophila* drives body shape change without molting[20], and elongation of wings and legs in *Drosophila* is controlled by ECM remodeling[21,22]. These findings suggest that body shaping during postembryonic development in insects depends not only on cuticle replacement but also on dynamic properties of the existing cuticle.

Here we demonstrate a "corset" property of the *Drosophila* larval cuticle that permits less efficient expansion of the cuticle along the body circumference than along the anteroposterior axis. During larval growth, this cuticle property directs a progressive change from a relatively round-shaped body to an elongated one. We further show that the corset property depends on the functions of two proteins that localize to a sub-surface layer of the larval cuticle, Cuticular protein 11 A (Cpr11A) and Tubby (Tb). The stretchable cuticle on which circumferential stiffness is conferred by its components allows the fly to execute postembryonic body elongation.

## Results

**Abnormal shaping of the larval cuticle in *Cpr11A* mutants**. We found that the pupae of *MB00532* homozygotes or hemizygotes, which harbor a transposon insertion immediately upstream of *Cpr11A* (Fig. 1a), appeared laterally wider and anteroposteriorly shorter than the wild-type pupae (Fig. 1c–f). The abnormal shapes of *MB00532* pupae were represented by their body axial ratios (length/width ratios), which were significantly smaller than those of the wild-type (Fig. 1k; compare +/+ with *MB/MB* and +/Y with *MB*/Y). *Df(1)Exel6244*, a deficiency uncovering *Cpr11A* along with six neighboring genes (Fig. 1a), failed to compliment the tubby phenotype of *MB00532* (Fig. 1g, k; compare *MB/Df* with *MB/MB*). *Df(1)Exel6244* hemizygotes were viable and exhibited even tubbier pupal shapes (represented by smaller axial ratios) compared with the *MB00532* hemizygotes (Fig. 1h, k; compare *Df*/Y with *MB*/Y). *Df(1)Exel6244* homozygous females were larval lethal. The tubby phenotype was recapitulated by whole-body expression of dsRNA against *Cpr11A* in the wild-type background (Fig. 1k; compare *Act*-GAL4 with *Act* > *Cpr11A* dsRNA). Conversely, the tubby phenotype of *MB00532* or *Df(1)Exel6244* was rescued by inducing whole-body expression of *Cpr11A* cDNA (Fig. 1i, k; *MB*/Y; *Act* > *Cpr11A*), or by introduction of *Cpr11A:EGFP*, a rescue construct containing the genomic

sequence of *Cpr11A* into which an EGFP-coding sequence was inserted immediately upstream of the *Cpr11A* stop codon (Fig. 1a, j, k; *Df*/Y; *Cpr11A:EGFP*/ + ). These results indicated that loss of *Cpr11A* function was responsible for the tubby body shape phenotype. *Df(1)Exel6244* hemizygotes are hereafter referred to as *Cpr11A^Df*.

*Cpr11A* encodes a protein harboring an N-terminal signal peptide and a Rebers and Riddiford (R&R) domain, a chitin-binding domain conserved in numerous arthropod cuticular proteins[23] (Fig. 1b). This suggested that *Cpr11A* might affect shaping of the cuticle. The third instar cuticle undergoes anteroposterior contraction and circumferential expansion at the onset of metamorphosis to directly become the puparium (pupal case) that covers the pupal body[20]. To test whether the pupal shape differences between the wild-type and the *Cpr11A* mutants reflect the effect of *Cpr11A* on larval cuticle shaping or on the metamorphic shape change, we isolated and made flat preparations of cuticles of wild-type, *Cpr11A^Df* and *MB00532* third instar larvae, and measured their axial ratios. We confirmed that the differences in pupal axial ratios mirrored the differences in the axial ratios of the larval cuticle per se (Fig. 2aVII, bVII, f, Supplementary Fig. 1). These results suggest that *Cpr11A* is required for normal shaping of the larval cuticle.

**Cpr11A-dependent anisotropy in cuticle expansion during larval growth**. The *Drosophila* larval period is divided into three instars by two molts, and lasts for approximately four days in total. During this period, the larva undergoes rapid and continuous increase in body weight (approximately 100-fold)[24–26], and the surface area of the cuticle increases in a continuous manner both between and across molts[18]. To investigate the origin of the cuticle shape abnormality of *Cpr11A* mutants, we analyzed the cuticle shapes of *Cpr11A^Df* larvae of various stages and body sizes. Siblings carrying the *FM7i, Act-GFP* balancer were used as controls.

The first instar larval cuticle is formed *de novo* during late embryogenesis, and is completed near the end of embryogenesis (stage 17d according to ref. [27].). The shapes of cuticles isolated at this stage were comparable between the control and the *Cpr11A^Df* mutant (Fig. 2aI, bI, dI), as represented by comparable axial ratios of the cuticle (Fig. 2e, f). After hatching, the cuticle widths and lengths increased in different proportions in the control and the mutant (Fig. 2d). Notably, the cuticle of the control larvae expanded anisotropically: it expanded at a higher rate along the length (body anteroposterior axis) than along the width (body circumference) (Fig. 2aI-VII), as represented by the progressive increase in the cuticle axial ratio (Fig. 2e, f). In contrast, cuticle expanded in a relatively isotropic manner in *Cpr11A^Df* larvae (Fig. 2bI-VII): the cuticle axial ratio remained fairly constant while the cuticle length increased (Fig. 2e), and changes in the cuticle axial ratio between stages were insignificant except for a small increase from the first instar to the second (Fig. 2e, f). The difference in anisotropy of cuticle expansion between the control and the mutant ultimately resulted in the marked difference in the body shapes of third instar larvae (Fig. 2aVII, bVII) and pupae (Fig. 1). These results indicate that, while *Cpr11A* does not visibly affect the initial shaping of larval cuticle during embryogenesis, it is required for the anisotropy of cuticle expansion during larval growth. Cpr11A may limit expansion of cuticle along the body circumference, and/or promote cuticle expansion along the anteroposterior axis, during body size increase.

**Comparison with the *Tubby^1* mutant phenotype**. The *Cpr11A* mutant phenotype was reminiscent of *Tubby^1* (*Tb^1*) and *TweedleD^1* (*TwdlD^1*) mutations, which individually cause tubby larval

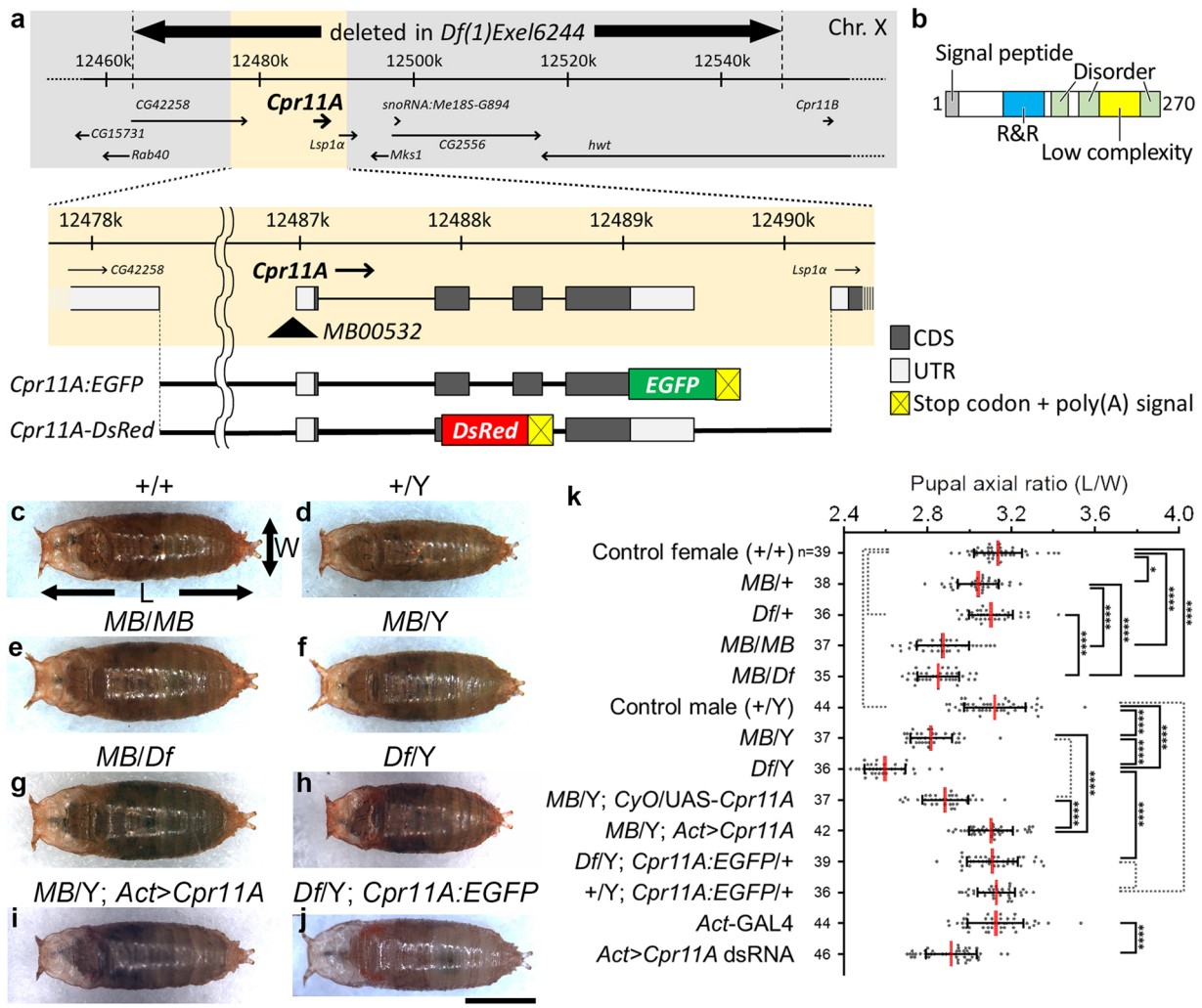

**Fig. 1 Pupal shape variations caused by modulations of Cpr11A function. a** Schematic representation of the *Cpr11A* locus (X chromosome), and mutations and transgenic constructs used in this study. The full-length Cpr11A protein fused at its C-terminus with EGFP is expressed from the *Cpr11A:EGFP* transgene. Nonsecretory DsRed is expressed from the *Cpr11A-DsRed* transgene. **b** Domain structure of Cpr11A. The signal peptide predicted by SignalP5.0[46], the R&R domain (PF00379) by Pfam[49], the low complexity region and the disorder regions by MobiDB[50] are shown. **c–j** Pupae of the indicated genotypes, viewed from the dorsal side. Anterior is to the left. The length (L) and the width (W) of a pupa were measured as indicated by arrows. *MB*, MB005322; *Df*, Df(1) Exel6244; *Act > Cpr11A*, *Act*-GAL4/UAS-Cpr11A; *Act > Cpr11A* dsRNA, *Act*-GAL4/UAS-Cpr11A dsRNA. Bar, 1 mm. **k** Mean ± S.D. of pupal axial ratios (L/W) of individual genotypes. n, the number of pupae measured for each genotype. Significance was assessed using one-way ANOVA ($p < 0.0001$) and Tukey's multiple comparisons test. $*p < 0.05$; $****p < 0.0001$; broken brackets, not significant.

and pupal shapes in a dominant manner. $Tb^1$ and $TwdlD^1$ are amino acid deletions in Tb/TweedleA and TwdlD, respectively, which are two of the Tweedle family of cuticular proteins[28]. The axial ratios of $Tb^1$ homozygotes (referred to as $Tb^1$ hereafter) cuticle isolated at embryonic stage 17d were comparable to those of the control and $Cpr11A^{Df}$ cuticle at the same stage (Fig. 2aI, bI, cI, d-f). During the first instar, the cuticle length-width plot of $Tb^1$ overlapped with that of $Cpr11A^{Df}$, and the cuticle axial ratios of $Tb^1$ and $Cpr11A^{Df}$ were comparable (Fig. 2aII-III, bII-III, cII-III, d-f). Then, the axial ratio of $Tb^1$ cuticle decreased significantly from the first instar to the second (Fig. 2e, f), representing a higher rate of cuticle expansion along the body circumference relative to expansion along the anteroposterior axis. The ratio further decreased from the second instar to the third (Fig. 2e, f). Thus, the $Tb^1$ mutation reduces or reverses the anisotropy of cuticle expansion during larval growth.

To test whether the *Cpr11A* and *Tb* mutations affected the growth of the larva, the ultimate body sizes of wild-type, $Cpr11A^{Df}$, MB00532 and $Tb^1$ larvae were assessed by weighing

male post-feeding (wandering) larvae. No statistically significant difference was found among them (Supplementary Fig. 2), indicating that the mutations did not affect the absolute body size.

**Direct measurement of cuticle stiffness.** Stretching of the larval cuticle due to rapid and continuous growth of the body volume has been proposed as one mechanism for cuticle expansion[18]. To test whether the difference in anisotropy of cuticle expansion between the control and the $Cpr11A^{Df}$ or $Tb^1$ mutants might reflect difference in how the cuticle responded to mechanical stretch, we directly applied circumferential or anteroposterior pulling force to the cuticle of third instar larvae through metal pads glued to the cuticle and examined how the cuticle extended in response (e.g. Figure 3a, see Methods). Figure 3b and d show how the cuticle strain, defined as increase in cuticle width/length divided by the initial width/length, changed in response to step-wise increasing stress, defined as pulling force applied per unit length/width of the cuticle. The strain-stress plots were roughly

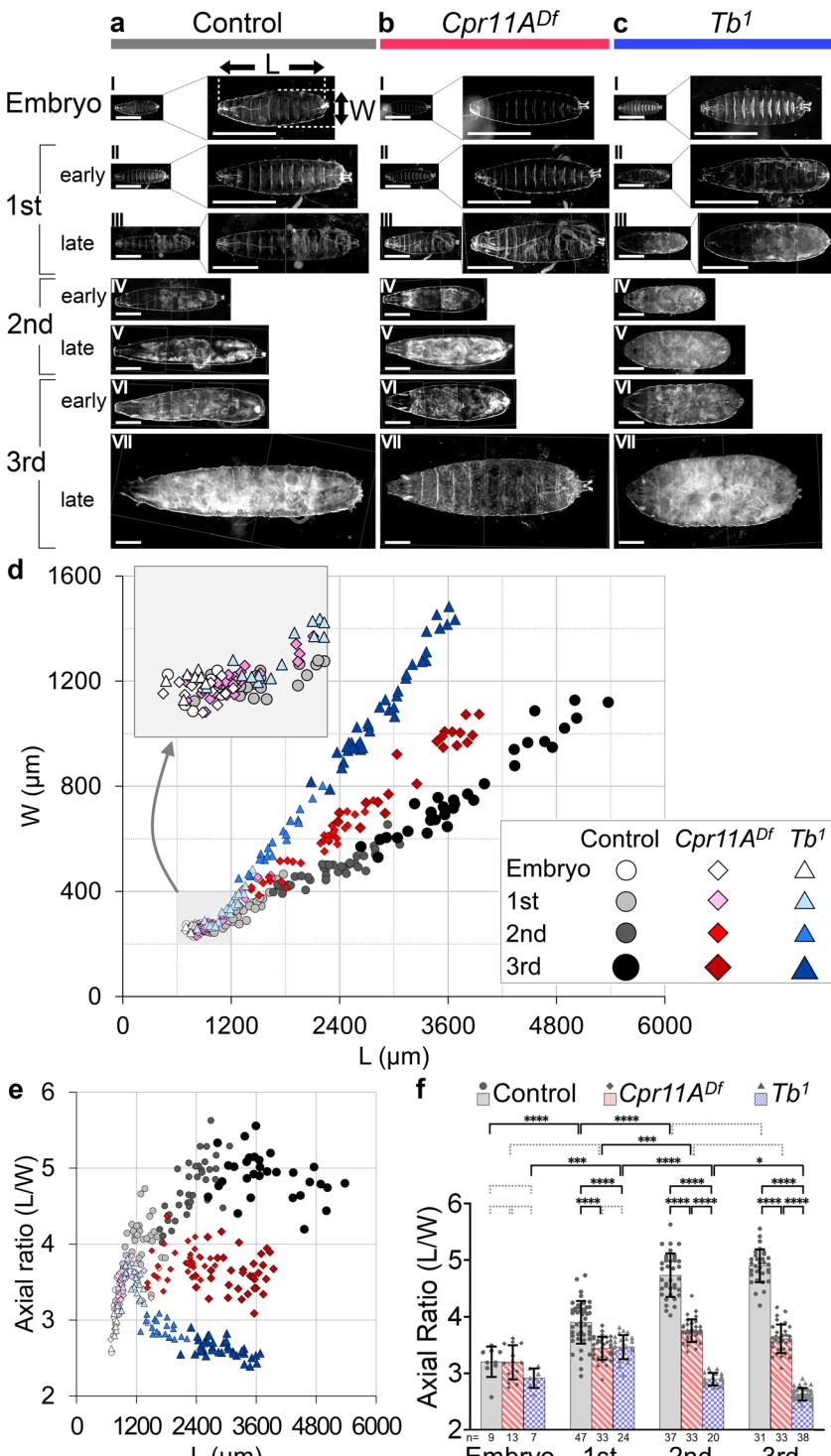

**Fig. 2 Cuticle expansion during larval growth. a–c** Larval cuticles of the control (**a**), $Cpr11A^{Df}$ (**b**) and $Tb^1$ (**c**) at embryonic stage 17d (I), early/late first instar (II/III), early/late second instar (IV/V), and early/late third instar (VI/VII). In (I-III), magnified images are shown on the right. Cuticles are viewed from the ventral sides. Anterior is to the left. L, cuticle length; W, cuticle width. Bars, 500 μm. **d** Plot of cuticle length (L) versus cuticle width (W). 1st, first instar; 2nd, second instar; 3rd, third instar. **e** Plot of cuticle axial ratio (L/W) versus cuticle length (L), calculated from the data in **d**. **f** Mean ± S.D. of axial ratios for each developmental stage, calculated from the data in (**d**). n, the number of embryos or larvae measured for each genotype for each stage. Significance was assessed using two-way ANOVA ($p < 0.0001$) and Tukey's multiple comparisons test. *$p < 0.05$; **$p < 0.01$; ***$p < 0.001$; ****$p < 0.0001$; broken brackets, not significant.

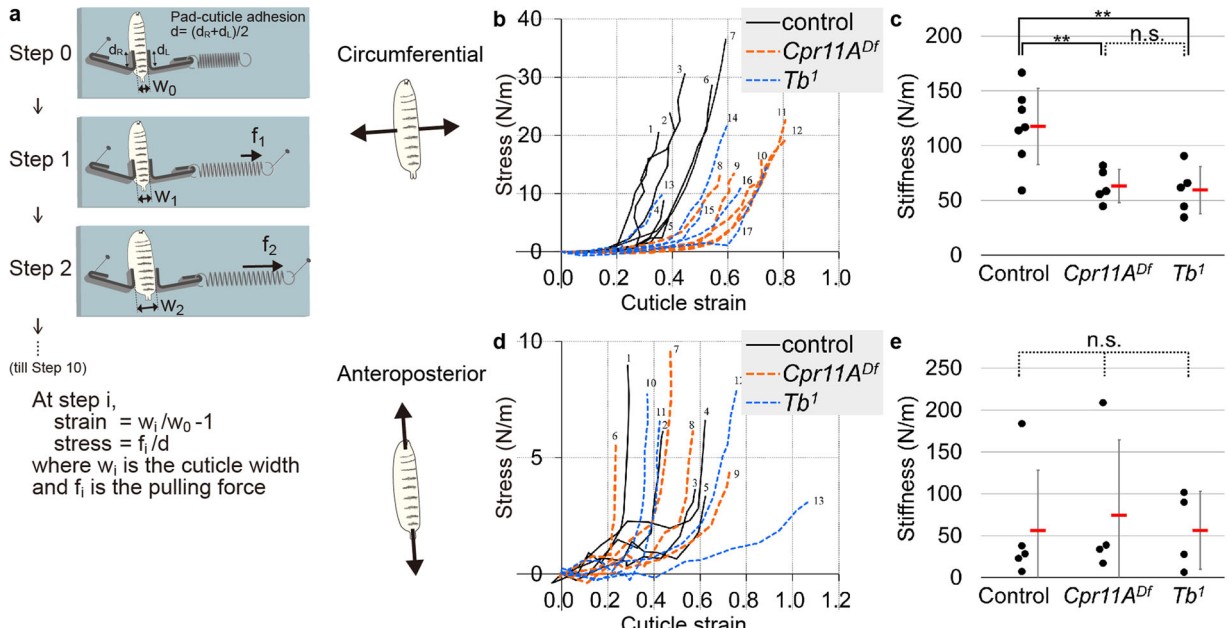

**Fig. 3 Direct measurement of cuticle stiffness. a–c** Measurement of cuticle stiffness along the body width (circumference). Numbers of individual larvae used for the measurement are 7 (control), 5 ($Cpr11A^{Df}$) and 5 ($Tb^1$). Significance was assessed using one-way ANOVA ($p = 0.0029$) and Tukey's multiple comparisons test. **$p < 0.01$; n.s., not significant. **d, e** Measurement of cuticle stiffness along the body anteroposterior axis. Numbers of individual larvae used for the measurement are 5 (control), 4 ($Cpr11A^{Df}$) and 4 ($Tb^1$). No significant difference in cuticle anteroposterior stiffness was found, as assessed by one-way ANOVA ($p = 0.914$). The larval identifiers in **b** and **d** (1, 2,...) correspond to the data in Supplementary Data.

linear in the area where stress was larger than 1 N/m, as indicated by high strain-stress correlation coefficients in the area (Supplementary Data). Stiffness was defined as the slope derived from linear regression of stress against strain in the area. (See Methods for details.)

While we did not find statistically significant differences in the anteroposterior stiffness (Fig. 3d, e), the circumferential cuticle stiffness of $Cpr11A^{Df}$ was significantly smaller than that of wild-type control cuticle (Fig. 3b, c). These results indicated that Cpr11A normally functions to limit the circumferential extension of cuticle. It should be noted that, although force application and measurement of extension were done on the cuticle, potential contributions from internal tissues such as the epidermis and the musculature were not eliminated, because the cuticles were not isolated in these measurements. Cpr11A may restrict circumferential cuticle expansion either directly, or indirectly through its effect on physical forces that the internal tissues exert upon the cuticle. The "corset" property conferred by Cpr11A, either directly or indirectly, is consistent with the $Cpr11A$-dependent anisotropy of cuticle expansion during larval growth; in the absence of Cpr11A function, the cuticle extends more easily along the circumference in response to internal tissue growth.

The circumferential cuticle stiffness of $Tb^1$ was significantly smaller than that of the normal cuticle and was comparable to that of $Cpr11A^{Df}$ (Fig. 3b, c). We did not find significant difference between $Cpr11A^{Df}$ and $Tb^1$ in cuticle stiffness, either along the body circumference or along the anteroposterior axis (Fig. 3b-e) (see Discussion).

**Expression and protein localization of Cpr11A**. Figure 4a shows the temporal profile of $Cpr11A$ mRNA expression extracted from modENCODE developmental transcriptome data[29]. $Cpr11A$ expression was up-regulated in the latter half of embryogenesis (when the synthesis of the first instar larval cuticle initiated[27]), persisted through first, second and early third instar larval stages, and decreased concomitantly with feeding cessation in the late

third instar. Thus, high levels of $Cpr11A$ expression occurred coincidently with larval cuticle synthesis and larval body growth.

According to in situ hybridization data of the Berkeley *Drosophila* Genome Project[30,31], $Cpr11A$ mRNA was expressed in the epidermis in a segmentally striped pattern in late embryos. To efficiently monitor gene expression and protein localization in the larva, we established a transgenic strain harboring two reporters, $Cpr11A$:EGFP and $Cpr11A$-DsRed (Fig. 1a). Functionality of the former in rescuing the body shape abnormality of the $Cpr11A^{Df}$ mutant as described above (Fig. 1i, j) suggested that the expression and the localization of the EGFP-tagged Cpr11A protein recapitulated those of the endogenous Cpr11A protein. The $Cpr11A$-DsRed reporter consisted of the $Cpr11A$ gene region in which the 5' part of the $Cpr11A$ coding sequence was replaced by a DsRed-encoding sequence followed by a stop codon and a polyadenylation signal, so that DsRed is not secreted and cells expressing the gene would be labeled by nonsecretory DsRed (Fig. 1a, see Methods). In first, second and third instar larvae, $Cpr11A$-DsRed and Cpr11A:GFP were expressed in a segmentally reiterated graded pattern, with high-level expression in the middle of every segment (Fig. 4b, c). In close-up views, DsRed signals were detected in the epidermal cells while the Cpr11A: EGFP protein was localized in the cuticle (Fig. 4d, cross-sections). When viewed from the surface, Cpr11A:EGFP was distributed throughout the cuticle and was enriched in a meshwork that transcended cell borders and continued over cells with much lower levels of $Cpr11A$-DsRed expression (Fig. 4d). The protein was also enriched in muscle attachment sites, despite lack of $Cpr11A$-DsRed expression in the underlying tendon cells (Fig. 4d). These observations suggest that the Cpr11A protein spreads and forms meshwork in the extracellular space.

**Protein interactions in a sub-surface region of the cuticle**. The insect cuticle is histologically subdivided into three layers: the envelop (the most apical (external) layer), the epicuticle (the medial layer consisting of proteins) and the procuticle (the basal

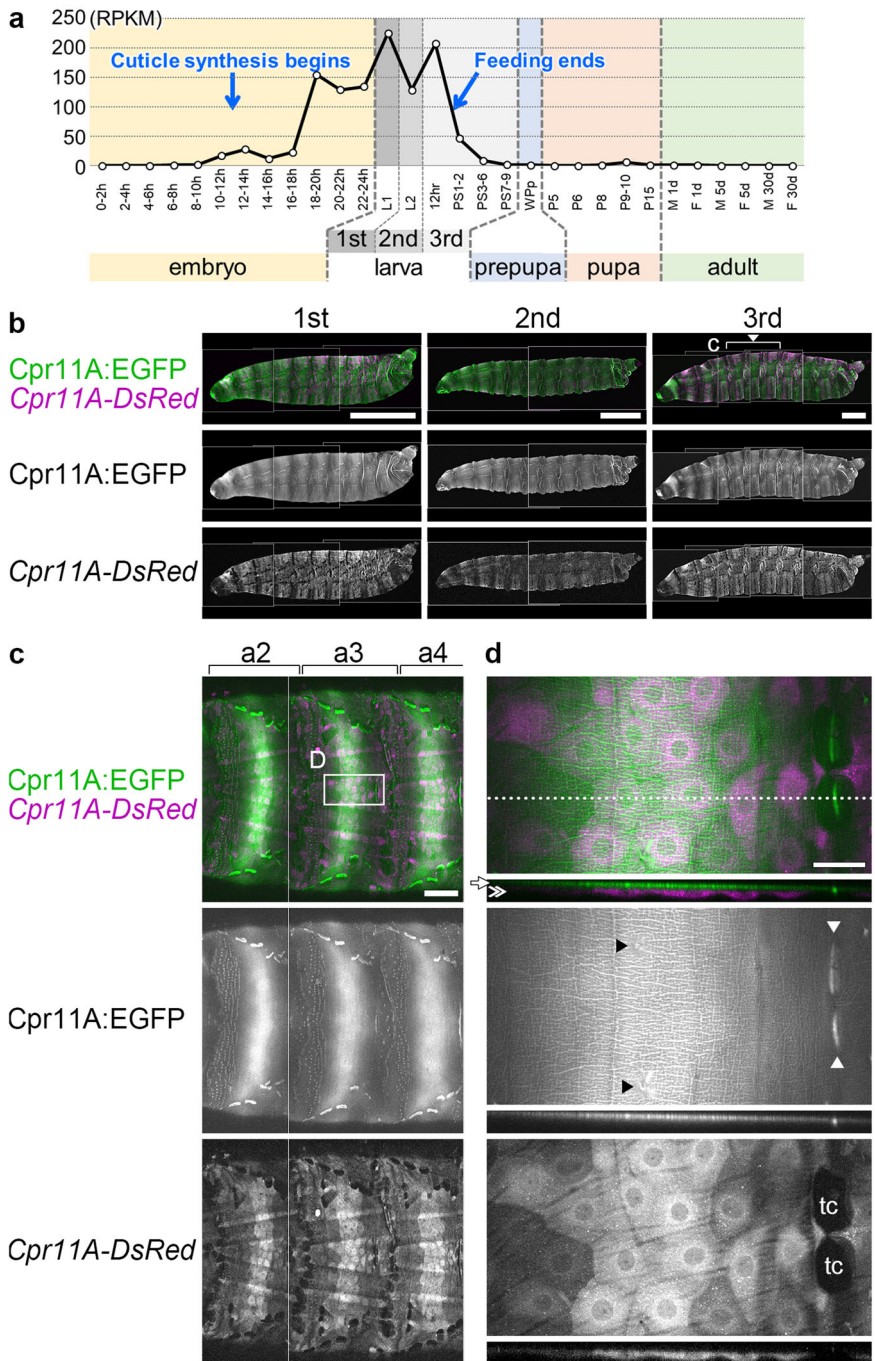

**Fig. 4 Cpr11A expression. a** The temporal RNA expression profile of *Cpr11A* extracted from modENCODE developmental transcriptome. h, hours after egg laying (AEL); 1st/L1, first instar larva; 2nd/L2, second instar larva; 3rd, third instar larva; 12 hr, 12 h post-molt (still feeding); PS1-2, dark blue gut stage (early wandering); PS3-6, light blue gut stage; PS7-9, clear gut stage; WPp, white prepupa, P, pupal stages; M, male; F, female; d, days after eclosion. The approximate timings of the beginning of first instar larval cuticle synthesis (embryonic stage 15[12], approximately 11:20-13 h AEL[27]) and the cessation of feeding (the beginning of wandering) are indicated. **b** EGFP and DsRed signals observed in live first, second and third instar larvae harboring Cpr11A:EGFP and *Cpr11A-DsRed* reporters. Larvae are viewed from the lateral sides. Bars, 500 μm. Anterior is to the left and dorsal is up. **c** The dorsal region of abdominal segments 2-4 (a2-4) (corresponding to the bracket region in **b**) of a third instar larva. Bar, 200 μm. **d** Upper panels are magnifications of the boxed region in **c**, and optical cross-sections at the position indicated by the broken line are shown in lower panels. **b**–**d** are projections of focal planes spanning the entire thickness of the cuticle and the epidermis. Apical (external) is up in cross-sections. Arrow, cuticle; double arrowhead, epidermal cells; tc, tendon cells; white arrowheads, muscle attachment sites in the cuticle; black arrowheads, sensory hairs. Bar, 50 μm.

layer consisting of proteins and chitin)[32]. The fluorescently tagged Tweedle proteins Tb:GFP[33] and TwdlD:DsRed[28] have been reported to mark an apical region of the larval cuticle. In our observation, the two proteins colocalized in a layer apical to the chitin-containing procuticle (Fig. 5a), and thus probably marked

the epicuticle. Cpr11A:EGFP formed a layer that spanned the basal portion of the TwdlD layer and the most apical portion of the procuticle (Fig. 5b-e), indicating that Cpr11A:EGFP was localized around the epicuticle-procuticle boundary. The apical side of Cpr11A:EGFP signals were flat while the basal side had

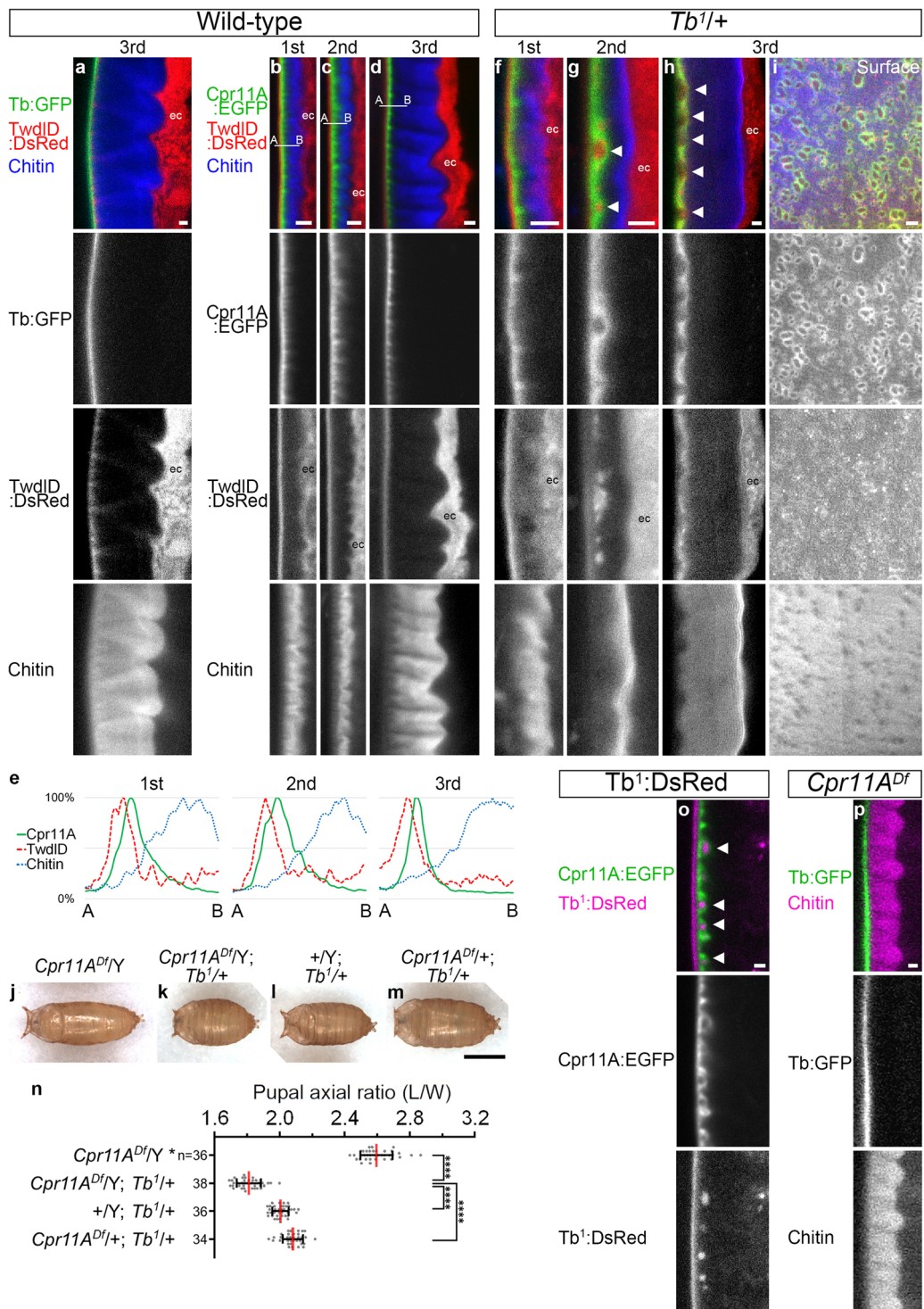

**Fig. 5 Interaction of Cpr11A with Twdl proteins. a** Localization of Tb:GFP, TwdlD:DsRed and chitin in a cross-section of a wild-type third instar larval cuticle. **b–d** Localization of Cpr11A:EGFP, TwdlD:DsRed and chitin in cross-sections of wild-type first (**b**), second (**c**) and third (**d**) instar larvae. **e** Profiles of Cpr11A:EGFP, TwdlD:DsRed and chitin signal intensities along the apical (A)-to-basal (B) lines shown in **b–d**, normalized by maximum signal intensity of each channel. **f–h** Cpr11A:EGFP, TwdlD:DsRed and chitin localization in cross-sections of $Tb^1/+$ first (**f**), second (**g**) and third (**h**) instar larvae. **i** Cpr11A: EGFP, TwdlD:DsRed and chitin localization viewed from the surface of a $Tb^1/+$ third instar larva. Anterior is to the left. Bar, 5 µm. **j–m** Pupae of the indicated genotypes, viewed from the dorsal side. Anterior is to the left. Bar, 1 mm. **n** Mean ± S.D. of pupal axial ratios (L/W) of individual genotypes. The same data for $Cpr11A^{Df}/Y$ used in Fig. 1j are used here for comparison. n, the number of pupae measured for each genotype. Significance was assessed using one-way ANOVA ($p < 0.0001$) and Tukey's multiple comparisons test. ****$p < 0.0001$. **o** Cpr11A:EGFP and Tb$^1$:DsRed localization in a cross-section of a $Cpr11A$: $EGFP/2xTb^1$:DsRed third instar larva. **p** Tb:GFP and chitin localization in a cross-section of a $Cpr11A^{Df}/Y$ third instar larva. In **a–d**, **f–h** and **o–p**, cross-sections are perpendicular to the anteroposterior axis, and apical (external) is to the left. Bars, 2 µm. Arrowheads, bubble-like basal bulges of Cpr11A:EGFP; ec, TwdlD:DsRed signals observed within epidermal cells.

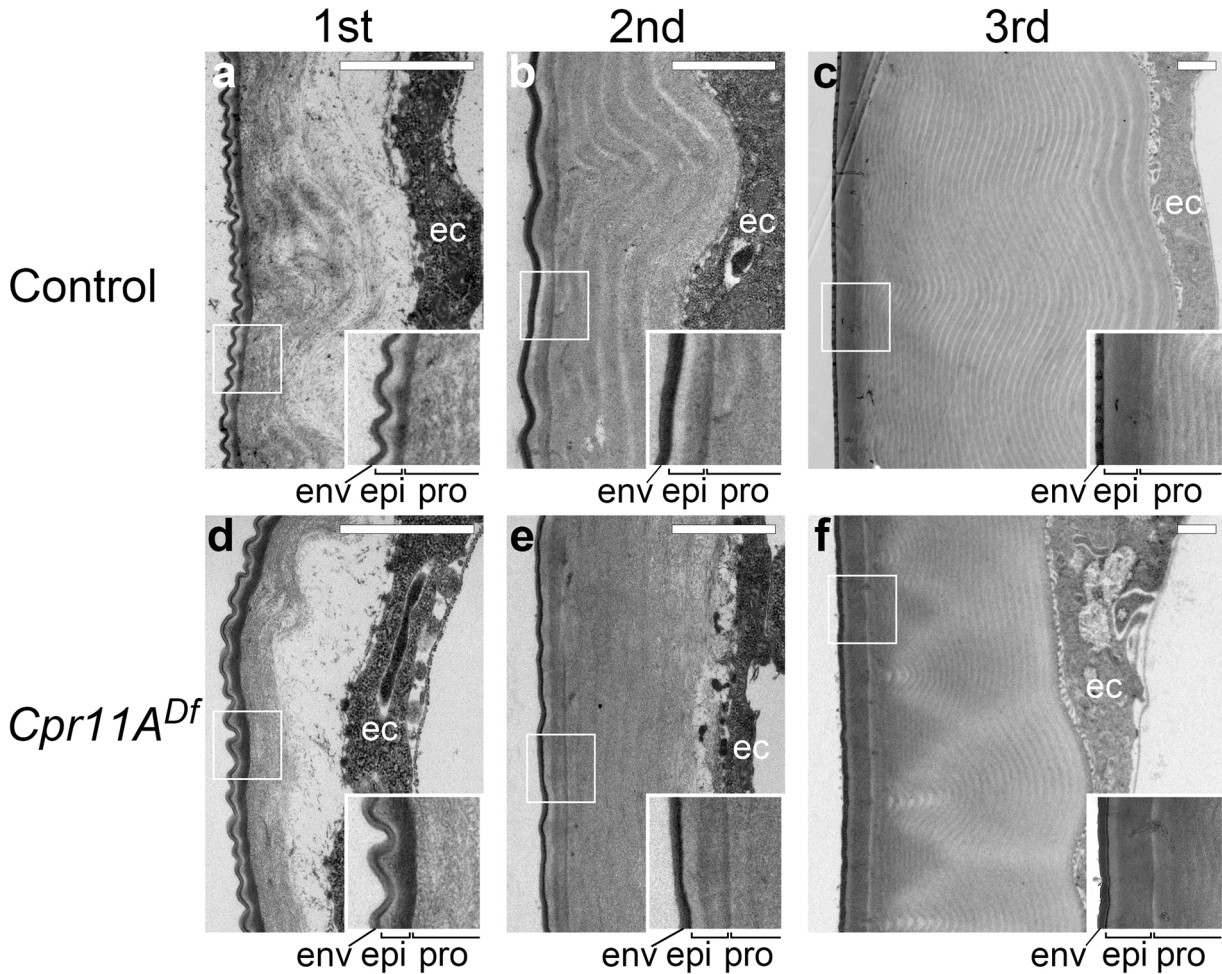

**Fig. 6 Cuticle ultrastructure.** Transmission electron micrographs of cuticle cross-sections perpendicular to the anteroposterior axis of the control (**a–c**) and the *Cpr11A^{Df}* mutant (**d–f**) in first (**a, d**), second (**b, e**) and third (**c, f**) instars. Apical (external) is to the left. ec, epidermal cells. Bars, 1 μm. Boxed regions are magnified on the right-lower corner of each panel. env, envelop; epi, epicuticle; pro, procuticle.

fine protrusions, which corresponded to the meshwork of Cpr11A:EGFP enrichment observed in surface views (Fig. 4d). Restricted localization of Cpr11A to the sub-surface region of the cuticle that is not in direct contact with the epidermis is suggestive of the protein's direct effect on cuticle mechanical property.

In the *Tb^1* mutant background, although Cpr11A:EGFP localization was normal during the first instar (Fig. 5f), a fraction of Cpr11A:EGFP formed bubble-like structures that contained TwdlD:DsRed and bulged basally during the second and the third instars (Fig. 5g, h).The remaining fraction localized to a sub-surface layer of the cuticle (Fig. 5g, h) but did not form a meshwork pattern (Fig. 5i, compare with Fig. 4d). Cpr11A:EGFP bubbles were also induced by the presence of the Tb^1:DsRed transgene, which expressed the DsRed-tagged mutant form of Tb and reproduced the *Tb^1* body shape phenotype[34]. The Cpr11A:EGFP bubbles contained the Tb^1:DsRed protein (Fig. 5o). The basal expansion of the epicuticle marker TwdlD:DsRed in the *Tb^1* background was reminiscent of abnormal epicuticle formation in the *Tb^1 Chubby/Kugel^v* double mutant, which was associated with accumulation of epicuticle materials in the procuticle[18]. We speculate that the Tb^1 protein singly interfered with normal epicuticle formation, which in turn affected Cpr11A localization.

In contrast, the localization of Tb:GFP was not visibly affected in the *Cpr11A^{Df}* mutant (Fig. 5p). Consistently, the cuticle ultrastructure of the *Cpr11A^{Df}* mutant, including the epicuticle-procuticle boundary, appeared normal (Fig. 6). Moreover, the localization of Tb:GFP and TwdlD:DsRed to the cuticle apical layer was invariable over areas with different levels of endogenous Cpr11A expression (Supplementary Fig. 3). Thus, the Cpr11A protein is likely to confer circumferential stiffness on the cuticle without affecting its overall organization.

To address whether the Cpr11A mislocalization was responsible for the abnormal cuticle shape of the *Tb^1* mutant, we tested whether the *Tb^1* mutation affected pupal shapes in the absence of *Cpr11A*. Interestingly, the *Tb^1* mutation caused significant difference in pupal axial ratios even in the absence of *Cpr11A* (Fig. 5j, k, n), suggesting that some corset property was retained even without Cpr11A. Loss of *Cpr11A* function in the *Tb^1* mutant background caused significant decrease in pupal axial ratios relative to the *Tb^1* single mutant (Fig. 5k-n), indicating that, despite partial mislocalization and disruption of the meshwork pattern, Cpr11A was still functional in the *Tb^1* mutant background.

**Extension of the epidermis.** We have shown that Cpr11A enables anisotropic cuticle extension through direct regulation of the cuticle mechanical property. Meanwhile, the epidermis continues to adhere to the cuticle within each instar, which necessitates anisotropic extension of the epidermis that matches the

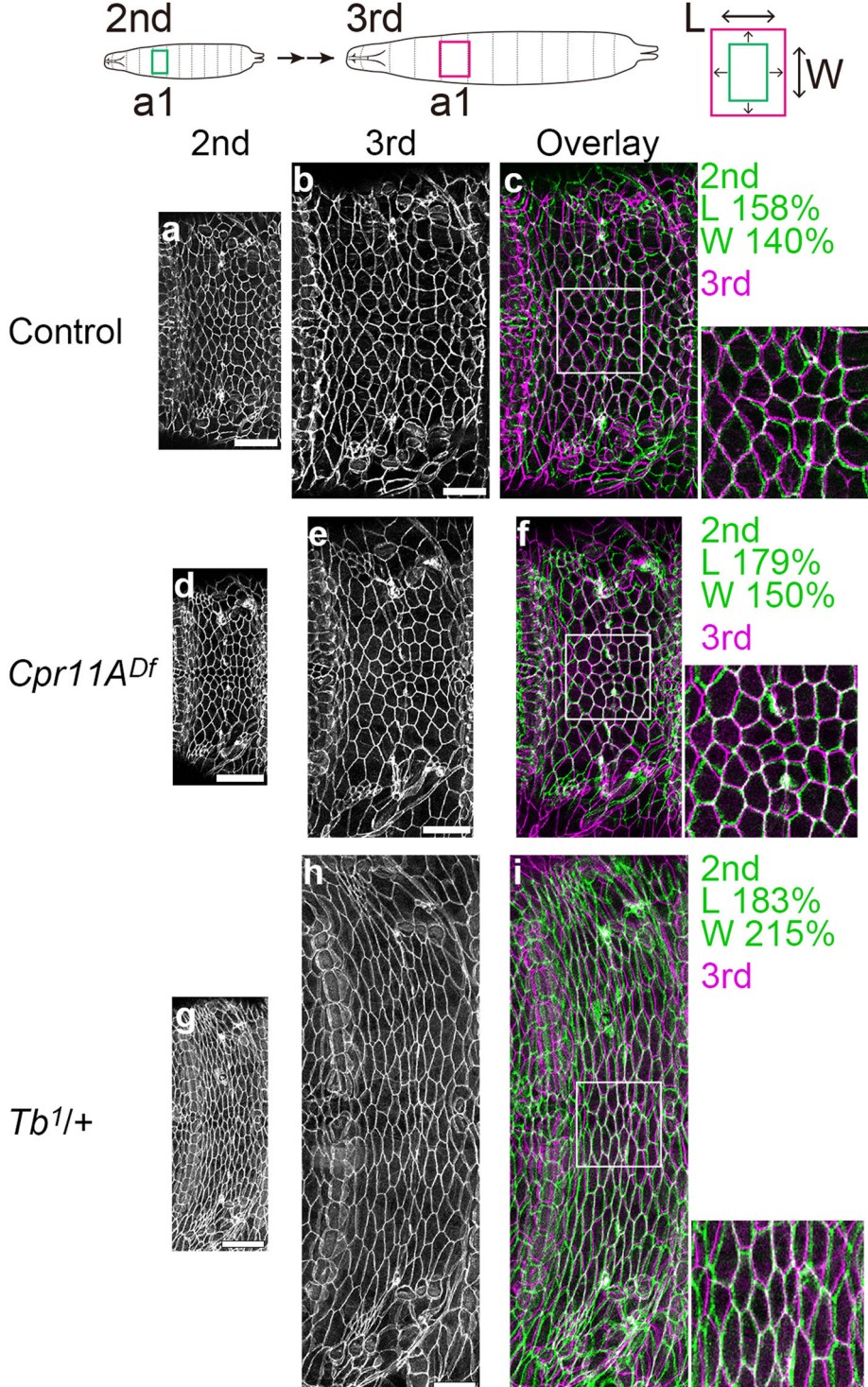

**Fig. 7 Epidermal cell outlines during larval growth.** Cell-cell boundaries in the epidermis, marked by Nrg:GFP, on the dorsal side of the abdominal segment 1 (a1) of the control (**a**, **b**), Cpr11A$^{Df}$ (**d**, **e**) and Tb$^1$/+ (**g**, **h**) larvae at the second instar (**a**, **d**, **g**) and two days later at the third instar (**b**, **e**, **h**). In **c**, **f** and **i**, the third instar images shown in magenta are overlaid on the second instar images enlarged along the body length (L) and width (W) by the indicated percentages and shown in green. Magnifications of the boxed regions are shown on the right. Bars, 50 μm.

cuticle extension. Tissue extension events have been accounted for by coordinated cell behaviors such as cell rearrangement and oriented cell division. On the other hand, the *Drosophila* larval epidermal cells are known to undergo size increase by endor-eplication throughout the larval period[35]. To examine cell dynamics in the epidermis during larval growth, we examined how the epidermal cell outlines (labeled with Neuroglian (Nrg):

GFP[36]) changed during growth, either across a molt or within a single instar, in live larvae of the wild-type, Cpr11A$^{Df}$ or Tb$^1$. In all three genotypes, the initial cell outlines were closely matched to those after growth by simply enlarging the initial images by different percentages along the body length and width (Fig. 7, Supplementary Fig. 4). This result indicates that active cell behaviors such as rearrangement and oriented division occur

rarely, if at all, in the epidermis during larval growth. Anisotropic extension of the epidermis could be either mechanically induced through adherence to the anisotropically extending cuticle, or oriented cell growth, or both. Within each genotype, the length-to-width ratios of epidermis extension within the third instar were not significantly different from those from the second instar to the third (Supplementary Fig. 4m, n). The continuous extension of the epidermis is in line with the linear increase in cuticle lengths and widths, both between and across molts, within each genotype (Fig. 2d). The epidermal cells in the $Tb^1$ mutant appeared highly anisometric (elongated along the body circumference; Fig. 7g, h, Supplementary Fig. 4i-l). It is consistent with the constantly circumference-biased extension of the epidermis in this mutant (Supplementary Fig. 4m).

**Effects of $Cpr11A^{Df}$ and $Tb^1$ mutations on adult shapes**. The $Tb^1$ mutation has been anecdotally reported to cause "tubby" body shape even in the adult[37]. To examine how $Cpr11A^{Df}$ and $Tb^1$ mutations affect the adult body shape, we measured the axial ratios in the abdomens and the thoraces (nota) of adult males. The $Tb^1$ mutant indeed showed lower axial ratios (meaning shorter and/or wider shape) in both the abdomen and the notum relative to the control (Supplementary Fig. 5a, c, d, f, g, h). In the $Cpr11A^{Df}$ mutant, the axial ratios of the abdomen were significantly lower than those of the control, although the axial ratios of the notum did not show significant difference (Supplementary Fig. 5a, b, d, e, g, h). As neither $Cpr11A$ nor $Tb$ is expressed at a substantial level during the pupal stage, in which the adult cuticles are formed (Fig. 4a and Supplementary Fig. 5i), they are unlikely to function directly within the adult cuticles. We speculate that the tubby puparia of the mutants mechanically confine the shapes of the adult bodies that form inside them.

## Discussion

We have shown that circumferential stiffness ("corset" property) of the *Drosophila* larval cuticle directs anisotropic cuticle expansion during body size increase by restricting circumferential cuticle stretching, which results in progressively more elongated shape of the whole body (Fig. 8a). Axis elongation has been commonly explained by cell behaviors such as oriented cell division, cell shape changes, cell intercalation and cell migration. Our finding demonstrates that the cuticle can play an instructive mechanical role in body elongation. The corset property depends on the function of protein constituents of the cuticle, Cpr11A and Tb, which show overlapping localization in a sub-surface region of the cuticle (Fig. 8a). Postembryonic survival in harsh

environments necessitates animals to support and protect their bodies with materials with greater strength than cells. Coating the body with a stretchable cuticle and equipping it with components that confer circumferential stiffness appears to be a potent strategy for conciliating active morphogenesis with the protective function of the cuticle. In the nematode *Caenorhabditis elegans*, a number of mutations in cuticle collagen genes are known to result in the "dumpy" phenotype (short and fat body shape)[38]. Regulation of cuticle physical properties by cuticle constituents may be a general mechanism for body shaping in diverse organisms bearing exoskeleton.

Generally, the cuticle remains adhered to the epidermis during its synthesis. During a molt, separation of the epidermis from the cuticle, initiation of new cuticle synthesis, and shedding of the old cuticle occur sequentially. The normal shape of the first instar cuticle of the $Cpr11A^{Df}$ and $Tb^1$ mutants before hatching (Fig. 2aI, II, bI, II, cI, II, d-f) is likely to mirror normal morphogenesis of the epidermis during embryogenesis. Within each instar, cuticle length and width increase continuously (Fig. 2d). Contrarily to some insects whose cuticle is initially formed in folds which later unfold to accommodate body growth[39], the fly larval cuticle does not show unfolding; it undergoes continuous expansion within an instar, both by stretching and by addition of new components[18]. Assuming that addition of new components to the cuticle contributes equivalently to anteroposterior and circumferential expansion of the cuticle, restriction of circumferential cuticle stretching by the Cpr11A- and Tb-dependent "corset" property would result in anisotropic cuticle expansion. Upon molting, cuticle lengths and widths, as well as differences in cuticle axial ratios between genotypes, are carried over from one instar to the next (Fig. 2d–f). Presumably, the shape of the old cuticle is transmitted to the new cuticle through the epidermis: when new cuticle starts to be synthesized, its shape conforms to the surface shape of the epidermis, which in turn mirrors the shape of the cuticle of the previous instar from which the epidermis has just separated.

Here let us simplify stretching of the cuticle within each instar as inflation of a tube-shaped balloon in response to increasing internal pressure. The thin-walled pressure vessel theory posits that circumferential stress acting on the tube wall (cuticle) is twice as large as longitudinal (anteroposterior) stress[40]. Consistently, a blown up tube-shaped rubber balloon expands more efficiently along its circumference than along its longitudinal axis (Fig. 8b). If the cuticle stiffness along the body circumference is equivalent to that along the anteroposterior axis, the cuticle would stretch at a higher rate along the circumference than along the anteroposterior axis, which may be the case for the $Tb^1$ mutant.

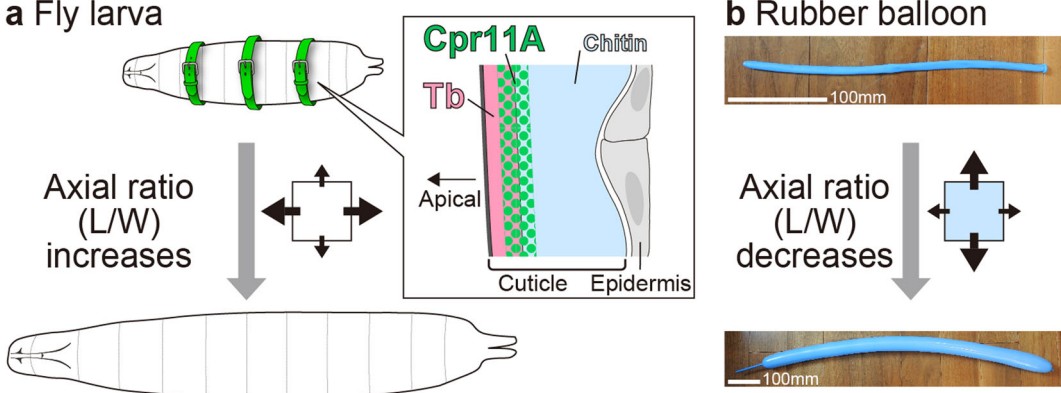

**Fig. 8 A corset function of the cuticle. a** A summary of this study. **b** A rubber balloon before (upper) and after (lower) inflation. In this case, the balloon expanded 3.2-fold along its longitudinal axis and 5.4-fold along its circumference.

The isotropic cuticle extension seen in the $Cpr11A^{Df}$ mutant would necessitate that cuticle be twice as stiff along the circumference as along the anteroposterior axis. If so, the Cpr11A protein in the wild-type larva would function to make the cuticle even stiffer along the circumference, thereby directing the cuticle to extend more preferentially along the anteroposterior axis. Cpr11A may confer circumferential stiffness directly on the cuticle, although, as noted above, the indirect action of Cpr11A through mechanics of internal tissues such as the epidermis and the musculature cannot be ruled out. We have not been able to explain the divergence in cuticle shaping between $Cpr11A^{Df}$ and $Tb^1$. The large variations in our measurement of the anteroposterior cuticle stiffness (Fig. 3e) might have masked potential difference between $Cpr11A^{Df}$ and $Tb^1$. The large variations also made it difficult to compare circumferential and anteroposterior stiffness values within individual genotypes.

Larval body elongation assisted by anisotropic cuticle expansion is reminiscent of the *Drosophila* egg elongation assisted by the surrounding ECM. Although it was initially proposed that the ECM itself acts as a molecular corset[41,42], it has recently been reported that F-actin within the egg acts as a molecular corset and that the ECM acts as a cue to polarize the F-actin[43]. In the case of larval body elongation, localization of Cpr11A and Tb to the sub-surface region of the cuticle that is not in direct contact with the epidermis suggests that those proteins directly confer the "corset" property on the cuticle. Nonetheless, the possibility that the proteins affect biophysical properties of the internal tissues, for example by acting as "signals" to the epidermis or the musculature, is not excluded. Future studies are needed to specify sites of action of those proteins.

Overlapping localization of Cpr11A, Tb and TwdlD, together with $Tb^1$-induced mislocalization of Cpr11A and TwdlD in or around the basal $Tb^1$ aggregates, suggests attractive interactions among those proteins. We speculate that those multiple interactions contribute additively to the corset property. The result that the $Cpr11A^{Df}$ and $Tb^1$ double mutant shows a more severe effect on cuticle shaping than either single mutant supports this model. Localization of Cpr11A and Tb to the sub-surface region of the cuticle is consistent with their effect on cuticle expansion and with the lack of effect on active cell behaviors in the epidermis. How are the protein localized to the cuticle region away from the cells from which they derive? One explanation would be that temporary deposition of those proteins by the epidermis prior to procuticle formation allows sequential stacking of the proteins and the procuticle. The temporal expression pattern of *Cpr11A* contradicts this scenario. For example, *Cpr11A* is expressed at high levels near the end of embryogenesis (18-24 h after egg laying (AEL), Fig. 4a), whereas the procuticle of the first instar cuticle is apparent in the embryo as early as 13-15 h AEL and has substantially thickened by 19 h AEL[27]. Likewise, *Cpr11A* expression is sustained at high levels for at least 12 h after the second molt (Fig. 4a), whereas the procuticle of the third instar cuticle is present before the molt and thickens continuously[18]. From the observation that the epicuticle thickness shows no decrease despite large increase in surface area during the second and the third instars, Kaznowski et al. inferred that the epidermis secretes material that moves through the procuticle to be added to the epicuticle[18]. Exclusive localization of the Cpr11A protein to the epicuticle-procuticle boundary despite its sustained production by the epidermis suggests that the protein penetrates through the procuticle to reach its destination. The R&R chitin-binding domain of Cpr11A may facilitate movement through the chitin-containing procuticle. Intracuticular movement of Cpr11A could also contribute to its planar spread into regions where the underlying cells do not express the *Cpr11A* gene (Fig. 4d). The attractive interaction between Cpr11A and the Twdl proteins discussed above, together with Cpr11A affinity to chitin, may stabilize Cpr11A around the epicuticle-procuticle boundary. We envision that cuticular protein interactions and dynamics that occur within the cuticle are of general significance in understanding cuticle organization and properties.

## Methods

**Fly stocks**. Flies were maintained on standard yeast-cornmeal food at 25 °C unless otherwise mentioned. Canton-S was used as wild-type. MB00532, Df(1)Exel6244, Tb1, FM7i Act-GFP, CyO 2xTb1-RFP (2xTb1:DsRed) and Actin5C-GAL4 (Act-GAL4) were obtained from the Bloomington Stock Center, and NrgG00305 (Nrg: GFP) was from the Kyoto Stock Center. v103321 obtained from the Vienna Drosophila Resource Center was used as UAS-Cpr11A dsRNA. Tb:GFP was a gift from B. Moussian and TwdlD:DsRed was a gift from S. Wasserman. Tb1 heterozygotes and homozygotes are morphologically inseparable[37]. To simplify mating schemes, we introduced transgenes and other mutations into the Tb1 heterozygous background in Figs. 5, 7 and Supplementary Fig. 4.

**Transgenic flies**. To generate the UAS-Cpr11A construct, we subcloned Cpr11A cDNA from RE57452 of *Drosophila* Gold Collection[44] into pUAST using EcoRI and KpnI. For generation of Cpr11A:EGFP and Cpr11A-DsRed constructs, pCaSpeR4-Cpr11A was established first by amplifying the 7.1 kb Cpr11A gene region flanked by the neighboring genes (Fig. 1a) from the Canton-S genome by PCR with primers [5'-AGG TCG ACC TCG AGG ACC AGT TCG CTG GAG AGA GA-3'] and [5'-AAC GTT AAC TCG AGG TGT CGT GGG TCT TGT GGT AA-3'], and cloning it into StuI-digested pCaSpeR4 using In-Fusion HD Cloning Kit (Clontech). The construct was then digested with BamHI. The region of the coding sequence removed by the BamHI digestion was amplified from the Canton-S genome with primers [5'-GCC AGA CCG TGG ATC CTA-3'] and [5'-GCC CTT GCT CAC CAT TGC CGT GGG CAG ATA CAG A-3']. A region of pPIG-A3GR[45] that contains the EGFP sequence, a stop codon and a polyadenylation signal was amplified using primers [5'-ATG GTG AGC AAG GGC GAG GA-3'] and [5'-TAG TGG ATC TGG ATC CCA GGT TCT TCA TTG GCT TC-3']. The two amplified fragments were inserted into the BamHI-digested construct by In-Fusion, to obtain Cpr11A:EGFP. For Cpr11A-DsRed, pCaSpeR4-Cpr11A was digested with BglII. The 1.8 kb genomic region starting from the BglII site within the upstream sequence and ending near the 5' end of exon 2 was amplified from the Canton-S genome using primers [5'-TTT GAG CCC AAG ATC TCA CG-3'] and [5'-TCT CGG AGG AGG CCA AAC CCA GAA GGA TCA GCT AA-3']. A region of pPIG-A3GR[25] that contains the DsRed sequence, a stop codon and a polyadenylation signal was amplified using primers [5'-TGG CCT CCT CCG AGA ACG T-3'] and [5'-AAG TTA TTA CAG ATC CAC TCA ACC CTA TCT CGG TC-3']. The two amplified fragments were inserted into the BglII-digested construct by In-Fusion, so that the protein to be expressed from the construct would consist of Cpr11A amino acids 1-9 (predicted by SignalP[46] to be nonfunctional as a signal peptide) fused in frame to DsRed. Transgenesis was done by BestGene. The transgenic strains are available from the corresponding authors upon request.

**Microscopy**. Larval cuticle and epidermis was fluorescently labeled and observed on the Olympus FV-1000 confocal laser microscope as described in[20]. Larval cuticle preparations and adult abdominal cuticle preparations were made according to[47] and[48], respectively, and were observed on the Olympus BX51 microscope. For transmission electron microscopy, ultrathin sections were made and were viewed on the JEOL JEM-1400 transmission electron microscope as described in[20].

**Measurement of cuticle elasticity**. For circumferential measurement, a planar metal pad (a half-bent staple) was glued to the right side of posterior cuticle of a third instar larva deeply anesthetized with diethyl ether vapor. The larva was placed on a rubber plate with its ventral side up, and the metal pad was fixed to the rubber plate with a pin. Another metal pad was glued to the left side of posterior cuticle, and was attached to one end of a tension spring (spring constant $= 5.1 \times 10^{-4}$ kgf/mm; Goko Hatsujo Co., Ltd.). The other end of the spring was pulled away in a stepwise manner (<1 mm per step, from step 0 (before pulling) to step 10) (Fig. 3a). The images of the cuticle and the spring were captured at each step on a Leica M165 FC stereo microscope, and the cuticle width and the spring length at each step were measured on ImageJ. We limited the placement of the metal pads (and thus the pulling) to the posterior part of the larval body so that increase in internal pressure arising from cuticle pulling would be released to the anterior part of the body. Circumferential cuticle strain at step i was defined as $w_i/w_0 - 1$, where $w_i$ is the cuticle width at step i and $w_0$ is the cuticle width before pull (step 0). Force applied to the spring at step i ($f_i$), calculated from the spring extension, was normalized by the average length of the pad-cuticle adhesion (d) to yield stress per cuticle unit length ($f_i/d$). As shown in Fig. 3b and d, the stress-strain plots were roughly linear in the area where stress >1 N/m, as indicated by high correlation coefficients (Supplementary Data). We defined circumferential stiffness of cuticle as the slope derived from linear regression of the stress-strain plot in this area. For anteroposterior elasticity measurement, metal pads were attached to anterior and posterior parts of cuticle on the ventral side, so that increased internal

pressure would be released to the dorsal side of the body. Subsequent procedures and data analyses were carried out as done for the circumferential measurement.

**Imaging of epidermal cell outlines.** Second instar or early third instar larvae expressing Nrg:GFP were anesthetized with diethyl ether vapor and mounted with water on glass slides. The epidermis of the dorsal side of the abdominal segment 1 was imaged by confocal laser microscopy, and the larvae were individually returned to fly media in 1.5 ml tubes. After incubation for one or two days at 25 °C, the epidermis of the same region was imaged again. In Fig. 7, Df(1)Exel6244, Nrg:GFP/Y are described as Cpr11A^Df. The progenies of crosses between Nrg:GFP females and either Canton-S males or Tb^1 males are described as control and Tb^1/+, respectively.

**Weighing larvae.** For each measurement, ten wandering larvae were collected and weighed on an electronic scale (Mettler Toledo AB104). 5-6 measurements were conducted per genotype.

**Statistics and reproducibility.** Data are presented as means ± S. D. The significance of differences between multiple groups were compared by one-way or two-way analysis of variance (ANOVA), followed by Tukey's multiple comparisons tests. Number of biological replicates are presented in figures or figure legends. Data were analyzed using GraphPad Prism 8.4.3 software. The number of animals examined for histology data in Figs. 4–6 and Supplementary Fig. 3 are presented in Supplementary Data file. At least 5 animals per stage per genotype were examined in fluorescent imaging and at least 3 animals per stage per genotype were examined in electron microscopy, and all animals gave similar results as shown in the figures.

**Reporting summary.** Further information on research design is available in the Nature Research Reporting Summary linked to this article.

## Data availability

All data supporting the findings of this study are included in this article. Source data for Figs. 1k, 2d-f, 3c, e, 5n and Supplementary Figs. 1, 2, 4m and 5g, h are provided in Supplementary Data 1.

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

## Acknowledgements

We thank the Kyoto Stock Center, the Bloomington Stock Center, Vienna *Drosophila* Resource Center, Bernard Moussian and Steven Wasserman for fly stocks, Nobuhiro Ogawa and Hiroaki Saito for use of TEM facility, and Shinya Komata for advice on statistical analyses. This work was supported by the Japan Society for the Promotion of Science (JSPS) Research Fellowships [15J40022 and 19J40012 to R.T.], JSPS Grants-in-Aid for Scientific Research [20K06666 to R.T., 20H00474 and 15H05778 to H.F., 18K06243 to T.K.], Grant-in-Aid for Scientific Research on Innovative Areas from the Ministry of Education, Culture, Sports, Science, and Technology (MEXT) [18H04758 to R.T.], the Naito Foundation (to R.T.), Tomizawa Jun-ichi & Keiko Fund of Molecular Biology Society of Japan for Young Scientist (to R.T.) and Shiseido Female Researcher Science Grant (to R.T.).

## Author contributions

R.T. and T.K. designed the experiments. R.T. performed the experiments. R.T., H.F. and T.K. analyzed the data. R.T. and T.K. wrote the manuscript.

## Competing interests

The authors declare no competing interests.
