## [Peer Review File · Communications Biology]

Reviewers' comments:

Reviewer #1 (Remarks to the Author):

This is an impressive manuscript that characterises the role of the exoskeleton as a 'corset' that drives elongation of the entire body during *Drosophila* larval development. Overall, the experiments are well performed and the results are significant and interesting. I have only one major comment that relates to the scholarship: why do the authors cite so few references? Many statements in the introduction section are made without any citations. And where citations are used, they are often in obscure journals (CMLS?) rather than citing more widely read reviews and articles on the same topic. It would be prudent for the authors to go through their manuscript line-by-line and add in multiple appropriate citations at each point. Otherwise it may appear that they are deliberately avoiding citing important papers on *Drosophila* morphogenesis.

For example, the concept of an extracellular ECM corset was proposed by the Bilder lab for the egg chamber, and there is a similar function in early wing disc metamorphosis, which might be a relevant example to cite here. There are also reviews on the role of cuticle in morphogenesis that contain relevant citations: <https://doi.org/10.1016/j.cois.2016.10.009>

Reviewer #2 (Remarks to the Author):

This manuscript studies how the shape of the *Drosophila* larvae and pupae is regulated. The authors reveal that the fly's larvae expand more efficiently in the AP axis than in the body circumference, which seems to be a critical developmental feature which is crucial to obtain the normal larval shape. The authors find that this mechanical property depends on the gene *Cpr11A*. These mutants show a similar phenotype to the one previously described in Tubby mutants.

The manuscript is well written, and the Figures are self-explanatory and very well presented.

Some issues need to be addressed, however, before the manuscript is ready for publication.

Specific comments below:

Are the *Cpr11A* mutants bigger in size than the controls? In addition to the defects in larval shape, those mutants appear to be bigger. It might be interesting to compare the size of those animals with control flies. This could suggest/reveal other potential functions of this protein.

Fig 1J: Why the "N" varies so much between the different genotypes studied? It ranges from 5 to 46. To present the data in a more consistent manner the authors should present a similar "N" between the different genotypes compared. This comment also applies to Fig 5N.

How is the adult shape in the *Cpr11A* mutants? Do they also show a similar phenotype as the one observed in *Tb* mutants?

Comment about the reference format in the main text: The first reference in the introduction is number 123; the next one is number 45; the following one, number 6; and so on... Shouldn't the refs follow a consecutive order starting from number 1?

Reviewer #3 (Remarks to the Author):

In this manuscript, the authors characterize the expression, localization and function of Cuticular protein 11A in maintaining the elongated shape of the larval body in *Drosophila*. They use modENCODE data to show that *Cpr11A* is expressed from late embryogenesis (during the period of

cuticle deposition) through middle/late 3rd instar, consistent with a role in continuous cuticle deposition as the organism grows. Using a GFP-tagged fully rescuing genomic construct and a DsRed non secretory reporter construct, the authors show that Cpr11A is expressed in segmentally repeated stripes in the larval epidermis, is secreted and localizes to the lower levels of the epicuticle. Loss of function alleles (homozygous and hemizygous *cpr11A* mutants and hemizygous *Df* animals) and animals expressing ubiquitous RNAi against *cpr11A* are viable and display a tubby phenotype in which the axial (L/W) ratio at the larval and pupal stage is significantly smaller than wild type control animals. This is similar, but less extreme, to the phenotype observed in *Tb1* animals. Animals hemizygous for *cpr11A* in a *Tb1* background show an even more tubby phenotype. Cpr11A localization in the epicuticle is disrupted in a *Tb1* background, whereas *Tb* expression and localization are unaffected in a *cpr11A* mutant background. Biophysical measurements of the larval cuticle along the anterior/posterior axis and circumferential axis reveal greater stiffness along the circumference, and that loss of Cpr11A and *Tb* result in reduced stiffness along this axis. Together, these results suggest that Cpr11A provides a mechanical stiffness property to the cuticle that constrains body shape along the circumference (a corset function), thereby allowing the body to preferentially elongate along the anterior/posterior axis during larval stages.

The experimental approaches are appropriate and well-controlled, and the results are novel and convincing. The manuscript is also well-organized and well-written. I believe that the subject would be generally interesting to the readers of Communications Biology. I do have a few concerns that should be addressed to improve this manuscript.

1. My biggest concern is that the entire manuscript treats the cuticle as a wholly independent structure, whereas it is physically attached to the epidermis and the shape of the organism is likely to be strongly influenced by physical constraints of the epidermis as well as the underlying musculature. This point needs to be considered in interpreting specific results and in the overall model. I will point out how this manifests itself in some of the subsequent points below.

2. Much of the initial phenotypic characterization and the rescue is performed using axial ratios from pupae. This is a little problematic as pupal shape is a reflection of muscular contraction of the larval animal followed by the hardening and tanning of the cuticle, and differs from the axial ratio in larvae (compare Figure 1 and 2). The authors did a nice job quantifying at least the control, *cpr11A* and *Tb1* mutant larvae, indicating that the axial ratio is also defective in the mutant larvae. It would be useful to acknowledge that axial ratios in pupae reflect both larval cuticle shape and the effects of metamorphosis and thus may not be truly indicative of the differences between the genotypes in the larvae.

3. The cell behavior experiment presented in Figure 7 is a bit rudimentary and underdeveloped. It appears to be one experiment for each animal and so is hard to evaluate. More importantly, the choice of times may miss important details. For example, in the larval trachea, the cuticle expands in the long axis with new cuticle added between taenidia during the intermolt period and then upon molting the spacing of taenidia reset with the formation of a new cuticle. It is certainly conceivable that there are more anisometric cell shape changes that happen between early and late in an instar and then reset upon molting. It would be more useful in this experiment to see how the cells and cuticle expand within a single instar. I would not be surprised if there are no cell rearrangements, but might expect to see larger differences in cell anisometries between early and late 2nd instar for example. It is furthermore difficult to ascribe whether any differences seen are the results of only the cuticle or of oriented cell growth (or a combination).

4. I don't believe the balloon analogy is very effective for a couple of reasons. First, and I may be wrong about this, the difference in axial ratio between uninflated and inflated 260 balloons has more to do with the initial shape of the balloon and not the properties of the material. A small round balloon made of the same material would inflate to a roughly isometric shape. Second, the cuticle is not the only thing opposing the internal pressure of the larva; radial expansion is also constrained by the epidermis. Although the cells could conceivably change shape from columnar to cuboidal or squamous, there are still cellular processes (junctional lengths, cytoskeleton, etc) that resist tissues remodeling in response to internal pressures.

5. It would be nice to show a figure of the protein domain structure of Cpr11A in Figure 1. This would also be helpful in setting up the experiments described in the expression and protein localization section of the results.

6. In the biophysical measurements, does circumferential pulling tear the cuticle from the epidermis?

7. The biophysical measurements of wild type and mutant larvae showed convincing differences along the circumferential axis, and supported the corset hypothesis. However, the experiment really demonstrated differences in stiffness of the "whole animal" rather than just the cuticle per se. If this were done on Hoyer's treated cuticle preparations rather than whole larvae it would be correct to make that claim, but these results also reflect contributions from the epidermis and musculature.

8. I'm curious how uniform the Cpr11A protein is in the epicuticle between segments. The photograph in Figure 4C makes it appear that it is strongly reduced a few cells away from the expressing cells. Is this a reflection of a confocal slice and the tissue dipping below the focal plane outside these cells, or is it really strongly reduced in these areas? If so, this could help explain the anisometric growth. Perhaps new cuticle is only added to areas outside of the Cpr11A zones. Note that this is how tracheal cuticle is added between taenidia during the intermolt period. This also raises questions about the ultrastructure. Are there any noticeable differences in cuticle structure (and localization of other cuticle markers such as Tb and TwdID) in Cpr11A high versus low portions of the cuticle?

9. There are a few places where the data fails to mention the number of animals examined and the penetrance of the phenotype. For example, how many animals were examined for the immunofluorescent staining of wild type and mutant cuticles and for the ultrastructural analysis? Were these phenotypes completely penetrant? Also, how many animals were examined for the cell outline study shown in Figure 7 and what is the variance on the L and W percentages for each to align?

Minor points:

a) Line 57 "some neighboring genes" is imprecise. Can you mention the number.

b) Line 75 I would argue that your results show that the overall body of the mutant animals are tubbier than wild type not just the cuticle.

c) I'm not sure that panel E in Figure 2 is that useful. The data presented in panels D and F is sufficient to make the point.

d) Line 288: "severer" should probably be replaced by "more severe."

e) Line 360: how are the larvae anesthetized?

First of all, we thank all reviewers for giving us thorough comments and suggestions, which helped us improve the manuscript. We have widened our data interpretation and discussion in the revised manuscript. We have also conducted additional experiments that were suggested by the reviewers, and described the results in the revised manuscript. Below, the comments from the reviewers are listed, and how we revised the manuscript accordingly is explained.

Reviewer #1 (Remarks to the Author):

This is an impressive manuscript that characterises the role of the exoskeleton as a 'corset' that drives elongation of the entire body during *Drosophila* larval development. Overall, the experiments are well performed and the results are significant and interesting. I have only one major comment that relates to the scholarship: why do the authors cite so few references? Many statements in the introduction section are made without any citations. And where citations are used, they are often in obscure journals (CMLS?) rather than citing more widely read reviews and articles on the same topic. It would be prudent for the authors to go through their manuscript line-by-line and add in multiple appropriate citations at each point. Otherwise it may appear that they are deliberately avoiding citing important papers on *Drosophila* morphogenesis.

For example, the concept of an extracellular ECM corset was proposed by the Bilder lab for the egg chamber, and there is a similar function in early wing disc metamorphosis, which might be a relevant example to cite here. There are also reviews on the role of cuticle in morphogenesis that contain relevant citations:

<https://doi.org/10.1016/j.cois.2016.10.009>

(Response)

We have added citations to major papers (references #1-5, 7, 9-10 on the reference list) and major textbooks (#11-13), including those suggested by the reviewer (#19, 21, 22), and have inserted text in the introduction section to refer to those papers (lines 35-36 “ECM dynamics has emerged as a general mechanism of insect morphogenesis “, lines 38-39 “elongation of wings and legs in *Drosophila* is controlled by ECM remodeling “). The role of egg chamber ECM in egg elongation is now cited (references #41-43) and is discussed in depth in the discussion section of the revised manuscript (lines 324-333).

Reviewer #2 (Remarks to the Author):

This manuscript studies how the shape of the *Drosophila* larvae and pupae is regulated. The authors reveal that the fly's larvae expand more efficiently in the AP axis than in the body circumference, which seems to be a critical developmental feature which is crucial to obtain the normal larval shape. The authors find that this mechanical property depends on the gene *Cpr11A*. These mutants show a similar phenotype to the one previously described in *Tubby* mutants.

The manuscript is well written, and the Figures are self explanatory and very well presented.

Some issues need to be addressed, however, before the manuscript is ready for publication.

Specific comments below:

Are the *Cpr11A* mutants bigger in size than the controls? In addition to the defects in larval shape, those mutants appear to be bigger. It might be interesting to compare the size of those animals with control flies. This could suggest/reveal other potential functions of this protein.

(Response)

To assess body sizes, we weighed male post-feeding (wandering) larvae of control, *Cpr11A*[Df], *Cpr11A*[MB00532] and *Tb*[1]. No statistically significant difference was found, suggesting that the mutations do not significantly affect the absolute body size. The result is shown in Supplementary Fig. 2 and is described in lines 129-132 of the revised manuscript.

Fig 1J: Why the "N" varies so much between the different genotypes studied? It ranges from 5 to 46. To present the data in a more consistent manner the authors should present a similar "N" between the different genotypes compared. This comment also applies to Fig 5N.

(Response)

We have conducted additional measurements for the genotypes that had insufficient sample numbers in the original manuscript. The net results are presented in the revised Fig 1K and 5N. Additional measurements do not affect the conclusions.

How is the adult shape in the *Cpr11A* mutants? Do they also show a similar phenotype as the one observed in *Tb* mutants?

(Response)

We have measured the axial ratios of the nota and the abdomens of wild-type, Cpr11A[Df] and Tb[1] adults. As presented in Supplementary Fig. 5, Cpr11A[Df] had a milder effect on the adult shape than Tb[1], similarly to its milder effect on the larval cuticle shape than Tb[1]. Absence of Cpr11A and Tb expression during the pupal stage argues against their direct functions within the adult cuticle. These results are discussed in the revised manuscript (lines 258-269).

Comment about the reference format in the main text: The first reference in the introduction is number 123; the next one is number 45; the following one, number 6; and so on... Shouldn't the refs follow a consecutive order starting from number 1?

(Response)

We had failed to put commas between reference numbers (for example, 123 should be 1, 2, 3). They are revised now.

Reviewer #3 (Remarks to the Author):

In this manuscript, the authors characterize the expression, localization and function of Cuticular protein 11A in maintaining the elongated shape of the larval body in *Drosophila*. They use modENCODE data to show that Cpr11A is expressed from late embryogenesis (during the period of cuticle deposition) through middle/late 3rd instar, consistent with a role in continuous cuticle deposition as the organism grows. Using a GFP-tagged fully rescuing genomic construct and a DsRed non secretory reporter construct, the authors show that Cpr11A is expressed in segmentally repeated stripes in the larval epidermis, is secreted and localizes to the lower levels of the epicuticle. Loss of function alleles (homozygous and hemizygous cpr11A mutants and hemizygous Df animals) and animals expressing ubiquitous RNAi against cpr11A are viable and display a tubby phenotype in which the axial (L/W) ratio at the larval and pupal stage is significantly smaller than wild type control animals. This is similar, but less extreme, to the phenotype observed in Tb1 animals. Animals hemizygous for cpr11A in a Tb1 background show an even more tubby phenotype. Cpr11A localization in the epicuticle is disrupted in a Tb1 background, whereas Tb expression and localization are unaffected in a cpr11A mutant background. Biophysical measurements of the larval cuticle along the anterior/posterior axis and circumferential axis reveal greater stiffness along the circumference, and that loss of Cpr11A and Tb result in reduced stiffness along this axis. Together, these results suggest that Cpr11A provides a mechanical stiffness property to the cuticle that constrains body shape along the circumference (a corset function), thereby allowing the body to preferentially elongate along the anterior/posterior axis during larval stages.

The experimental approaches are appropriate and well-controlled, and the results are novel and convincing. The manuscript is also well-organized and well-written. I believe that the subject would be generally interesting to the readers of *Communications Biology*. I do have a few concerns that should be addressed to improve this manuscript.

1. My biggest concern is that the entire manuscript treats the cuticle as a wholly independent structure, whereas it is physically attached to the epidermis and the shape of the organism is likely to be strongly influenced by physical constraints of the epidermis as well as the underlying musculature. This point needs to be considered in interpreting specific results and in the overall model. I will point out how this manifests itself in some of the subsequent points below.

(Response)

As the reviewer points out, contribution of other tissues such as the epidermis and the musculature on the body shape should not be neglected. We need to be particularly careful about this point in interpreting the results of the biophysical pulling (stress-strain) experiment, because we used the whole body, not just the cuticle, in this experiment. The revision that we have made on this issue is explained below in response to comment #7. Meanwhile, the cuticle preparations in Fig. 2A-C show the shape of the cuticle. The preparations were made by pricking and incubating larvae in glycerol:acetic acid (4:1) at 60°C, and were mounted flat in lactic acid:water (3:1). As a result, the internal tissues were killed and squashed, so that they do not contribute to the contour shape of the preparations, although some debris of internal tissues remain. Therefore, Fig. 2 does show that the cuticle itself expands anisotropically in the wild-type, and that the normal functions of Cpr11A and Tb are surely required for restricting circumferential expansion of the cuticle. Again, whether the restriction of circumferential cuticle expansion is driven solely by the physical property of the cuticle itself or is contributed by other tissues remains to be addressed in future studies. We have added discussion on this issue in the revised manuscript (lines 324-333).

2. Much of the initial phenotypic characterization and the rescue is performed using axial ratios from pupae. This is a little problematic as pupal shape is a reflection of muscular contraction of the larval animal followed by the hardening and tanning of the cuticle, and differs from the axial ratio in larvae (compare Figure 1 and 2). The authors did a nice job quantifying at least the control, cpr11A and Tb1 mutant larvae, indicating that the axial ratio is also defective in the mutant larvae. It would be useful to acknowledge that axial ratios in pupae reflect both larval cuticle shape and the effects of metamorphosis and thus may not be truly indicative of the differences between the genotypes in the

larvae.

(Response)

Indeed, we need to take into account the dramatic change in body shape that occurs during pupariation. This point is described in the revised manuscript (lines 77-79). We have compared the larval cuticle shapes and the pupal shapes of wild-type and two Cpr11A mutants (Df and MB00532), and confirmed that the pupal shape differences reflected the differences in the larval cuticle shapes. The comparison is presented in Supplementary Fig. 1 and is described in lines 79-87 of the revised manuscript.

3. The cell behavior experiment presented in Figure 7 is a bit rudimentary and underdeveloped. It appears to be one experiment for each animal and so is hard to evaluate. More importantly, the choice of times may miss important details. For example, in the larval trachea, the cuticle expands in the long axis with new cuticle added between taenidia during the intermolt period and then upon molting the spacing of taenidia reset with the formation of a new cuticle. It is certainly conceivable that there are more anisometric cell shape changes that happen between early and late in an instar and then reset upon molting. It would be more useful in this experiment to see how the cells and cuticle expand within a single instar. I would not be surprised if there are no cell rearrangements, but might expect to see larger differences in cell anisometries between early and late 2nd instar for example. It is furthermore difficult to ascribe whether any differences seen are the results of only the cuticle or of oriented cell growth (or a combination).

(Response)

In the initially submitted manuscript, comparison of cell outlines between the second and the third instars was conducted for 7-8 individuals per genotype, and absence of cell rearrangement and division was confirmed in all cases. We have added presentation of examples other than those presented in Fig 7 as Supplementary Fig. 4 A, B, E, F, I, J, and sample numbers are now shown in Supplementary Fig. 4N. Additionally, we have conducted comparison between early and late third instar for 7-9 individuals per genotype. We have calculated the rate of epidermis extension along the body length and width, and have found no significant difference between intra-instar and inter-instar changes. These results are presented in Supplementary Fig. 4 and described in the revised manuscript (lines 252-256).

As the reviewer points out, oriented cell growth may contribute to the anisotropic extension. We have added this possibility in the revised manuscript (lines 250-252 “Anisotropic extension of the epidermis could be either mechanically induced through adherence to the anisotropically extending cuticle, or oriented cell growth, or both.”).

4. I don't believe the balloon analogy is very effective for a couple of reasons. First, and I may be

wrong about this, the difference in axial ratio between uninflated and inflated 260 balloons has more to do with the initial shape of the balloon and not the properties of the material. A small round balloon made of the same material would inflate to a roughly isometric shape. Second, the cuticle is not the only thing opposing the internal pressure of the larva; radial expansion is also constrained by the epidermis. Although the cells could conceivably change shape from columnar to cuboidal or squamous, there are still cellular processes (junctional lengths, cytoskeleton, etc) that resist tissues remodeling in response to internal pressures.

(Response)

In the case of a symmetrical sphere, stress on the wall of a pressurized sphere is symmetrical so that a round balloon would indeed inflate isometrically. The balloon which we are discussing in Fig. 8 as analogous to the larva is tube (vessel)-shaped. According to a thin-walled pressurized vessel theory in physics, circumferential stress acting on the vessel wall is twice as large as longitudinal stress, regardless of the wall material. How the wall responds to the stresses would depend on the properties of the wall material, which we propose here to differ between the control and the mutants. This “wall material”, as pointed out in the second point, includes other tissues such as the epidermis and the musculature, which may also contribute to restricting circumferential expansion. In the revised manuscript, we treat the tube balloon analogy as a simplified scenario (line 305 “let us simplify ...”, line 314 “If so, ...”), and we have added consideration of other tissues contributing to circumferential stiffness (lines 317-319 “as noted above, the indirect action of Cpr11A through mechanics of internal tissues such as the epidermis and the musculature cannot be ruled out”).

5. It would be nice to show a figure of the protein domain structure of Cpr11A in Figure 1. This would also be helpful in setting up the experiments described in the expression and protein localization section of the results.

(Response)

The domain structure of the Cpr11A protein is now presented in Fig 1B.

6. In the biophysical measurements, does circumferential pulling tear the cuticle from the epidermis?

(Response)

No, it does not result in cuticle-epidermis separation. The potential contribution of tissues other than the cuticle on this measurement is addressed in the revised manuscript, as described below in response to comment #7.

7. The biophysical measurements of wild type and mutant larvae showed convincing differences along the circumferential axis, and supported the corset hypothesis. However, the experiment really demonstrated differences in stiffness of the “whole animal” rather than just the cuticle per se. If this were done on Hoyer’s treated cuticle preparations rather than whole larvae it would be correct to make that claim, but these results also reflect contributions from the epidermis and musculature.

(Response)

As the reviewer points out, contributions from other tissues such as the epidermis and the musculature are not eliminated in these measurements. We have clarified this point in the revised manuscript (lines 152-155 “It should be noted that, although force application and measurement of extension were done on the cuticle, potential contributions from internal tissues such as the epidermis and the musculature were not eliminated, because the cuticles were not isolated in these measurements”), and have revised the interpretation (lines 155-157 “Cpr11A may restrict circumferential cuticle expansion either directly, or indirectly through its effect on physical forces that the internal tissues exert upon the cuticle.”). As mentioned above in response to comments #1 and #4, the discussion section is also extensively revised to include consideration of other tissues (lines 316-319 and lines 324-333).

8. I’m curious how uniform the Cpr11A protein is in the epicuticle between segments. The photograph in Figure 4C makes it appear that it is strongly reduced a few cells away from the expressing cells. Is this a reflection of a confocal slice and the tissue dipping below the focal plane outside these cells, or is it really strongly reduced in these areas? If so, this could help explain the anisometric growth. Perhaps new cuticle is only added to areas outside of the Cpr11A zones. Note that this is how tracheal cuticle is added between taenidia during the intermolt period. This also raises questions about the ultrastructure. Are there any noticeable differences in cuticle structure (and localization of other cuticle markers such as Tb and Twd1D) in Cpr11A high versus low portions of the cuticle?

(Response)

Fig. 4B-D show projections of multiple focal planes spanning the thickness of the cuticle and the epidermis, so there is actual difference in Cpr11A level along the anteroposterior axis. In the revised manuscript, we have clarified that the images are projections (lines 628-629). We do not know whether the difference in Cpr11A level means differential rate of cuticle synthesis: we have not directly tested whether the rate of cuticle expansion differs in Cpr11A-high and -low areas. Fig. 7 and Supplementary Fig. 4 show epidermal cell outlines in the entire a1 segment (=containing both Cpr11A-high and -low areas). Initial and post-growth cell outlines are matched by uniform enlargement of initial images, suggesting that the epidermis extended at a comparable ratio in Cpr11A-high and -low areas. This

result rather argues against the possibility of spatially restricted extension. Taken together, we find it premature to discuss for or against the possibility that the spatially varied Cpr11A level may contribute to anisotropic expansion.

Regarding the cuticle structure, we have added data on the spatial distribution of Tb:GFP and Twd1D:DsRed in Supplementary Fig. 3, which show an area equivalent to the one shown in Fig. 4D (as marked by the position of sensory hairs (black arrowheads)). The localization of Tb:GFP and Twd1D:DsRed to the cuticle apical layer was invariable in Cpr11A-high and -low areas, indicating that the difference in Cpr11A level did not affect the apical-basal organization of the cuticle. The result is described in the revised manuscript (lines 222-224 “the localization of Tb:GFP and Twd1D:DsRed to the cuticle apical layer was invariable over areas with different levels of endogenous Cpr11A expression”). This is consistent with the normal localization of Tb:GFP in the Cpr11A[Df] mutant, and normal cuticle ultrastructure in the mutant.

Regarding the cuticle ultrastructure in Cpr11A-high versus -low areas of a wild-type larva: in conducting transmission electron microscopy shown in Fig. 6, we made sure to observe ultrathin sections in areas that did not contain dorsal hairs. Dorsal hairs are located within Cpr11A-low areas (the dot-like autofluorescence signals in the Cpr11A-low areas in Fig.4C represent dorsal hairs). As it is technically difficult to specify at a finer scale the position of an ultrathin section along the anteroposterior axis, we consider it unrealistic to precisely select ultrathin sections of Cpr11A-high versus -low areas. In addition, the use of the electron microscopy facility that we had been using (which is located in an institute outside of our laboratory) is restricted under the current situation of Covid-19. For these reasons we have chosen to focus on analyzing Tb and Twd1D localization. As described above, we believe that the result shown in Supplementary Fig. 3 has addressed the reviewer’s concern.

9. There are a few places where the data fails to mention the number of animals examined and the penetrance of the phenotype. For example, how many animals were examined for the immunofluorescent staining of wild type and mutant cuticles and for the ultrastructural analysis? Were these phenotypes completely penetrant? Also, how many animals were examined for the cell outline study shown in Figure 7 and what is the variance on the L and W percentages for each to align?

(Response)

The number of animals examined for histology data in Figs 4-6 and Supplementary Fig. 3 are now presented in Supplementary Data file. At least 5 animals per stage per genotype were examined in fluorescent imaging and at least 3 animals per stage per genotype were examined in electron microscopy, and the results shown in the figures were reproduced in all animals examined. We have added “Statistics and Reproducibility” in the Methods section to include these points (lines 442-450

of the revised manuscript).

In the cell outline study, the L and W percentages depend on the body size of the larva at the initial observation and how much it grows until the second observation. The L and W percentages are linearly correlated, as shown in Supplementary Fig. 4M-N. As mentioned above, the results of linear regression is discussed in lines 252-256 of the revised manuscript.

Minor points:

a) Line 57 “some neighboring genes” is imprecise. Can you mention the number.

(Response)

The number is mentioned in the revised manuscript (line 59 “six neighboring genes”).

b) Line 75 I would argue that your results show that the overall body of the mutant animals are tubbier than wild type not just the cuticle.

(Response)

As explained above in response to comment #1, the cuticle preparations in Fig. 2A-C show the shape of the cuticle. Internal tissues are killed and squashed, so they do not contribute to the contour shape of the preparations.

c) I’m not sure that panel E in Figure 2 is that useful. The data presented in panels D and F is sufficient to make the point.

(Response)

Yes, Fig. 2E is just a paraphrase of Fig. 2D, and essentially argues the same point as Fig. 2F does. However, just to clarify that the axial ratio of Cpr11A[Df] stays constant while L increases, we wish to leave this panel here if space allows.

d) Line 288: “severer” should probably be replaced by “more severe.”

(Response)

The replacement is made in the revised manuscript (line 338 “more severe effect”).

e) Line 360: how are the larvae anesthetized?

(Response)

Larvae were anesthetized with diethyl ether vapor. Method description is added in the revised manuscript (lines 408-409 and 430-431).

Below are updated figures.

Fig. 1

Changes:

Panel B (domain structure of Cpr11A) is added.

Additional measurements are included in panel K.

Fig. 4

Change:

Black arrowheads indicating sensory hairs are added in panel D, to mark equivalent positions in Supplementary Fig. 3.

Fig. 5

Change:

Additional measurements are included in panel N.

REVIEWERS' COMMENTS:

Reviewer #2 (Remarks to the Author):

The authors have addressed all my concerns and the paper is ready for publication in its current version.

Reviewer #3 (Remarks to the Author):

I am satisfied with the response to my previous review of this manuscript. The revised manuscript addresses all of my concerns to the extents possible, and is very balanced in indicating that underlying tissue dynamics may contribute to these phenotypes. I have two issues for the authors to consider. First, I am a bit confused about the Tb1 mutations used in the study. For the larval phenotypic analysis it appears that the Tb1 homozygotes were used, but then for the immunofluorescence and the pupal phenotypes, it was a heterozygote. Is there a reason for the switch in the genotypes and are there no differences in the larvae between the two? My second question relates to the anisometric cell shapes in Tb1 versus the control and Cpr11A larvae. Now that you have included more images in the supplemental figures, this difference seems really strong. You might consider pointing that out in the discussion either around lines 312 or 322.

Below are the comments from reviewer #3 and how we revised the manuscript accordingly.

“I am satisfied with the response to my previous review of this manuscript. The revised manuscript addresses all of my concerns to the extents possible, and is very balanced in indicating that underlying tissue dynamics may contribute to these phenotypes. I have two issues for the authors to consider. First, I am a bit confused about the Tb1 mutations used in the study. For the larval phenotypic analysis it appears that the Tb1 homozygotes were used, but then for the immunofluorescence and the pupal phenotypes, it was a heterozygote. Is there a reason for the switch in the genotypes and are there no differences in the larvae between the two?”

Tb[1] is a dominant mutation, and heterozygotes and homozygotes are morphologically inseparable. We used the Tb[1] heterozygous background in Figs. 5, 7 and Supplementary Fig. 4 because it made the mating schemes easier. These points are described in the Methods section of our revised manuscript (lines 380-383).

“My second question relates to the anisometric cell shapes in Tb1 versus the control and Cpr11A larvae. Now that you have included more images in the supplemental figures, this difference seems really strong. You might consider pointing that out in the discussion either around lines 312 or 322.

We have added a text pointing out the anisometric cell shapes of Tb[1] in the revised manuscript (lines 264-267). The reviewer suggested addition in the Discussion section, but we find that this description fits better where we placed it.